# Effect of Packaging and Salt Content and Type on Antioxidant and ACE-Inhibitory Activities in Requeson Cheese

**DOI:** 10.3390/foods11091264

**Published:** 2022-04-27

**Authors:** Ivette Karina Ramírez-Rivas, Néstor Gutiérrez-Méndez, Ana Luisa Rentería-Monterrubio, Rogelio Sánchez-Vega, Juan Manuel Tirado-Gallegos, Eduardo Santellano-Estrada, Martha María Arevalos-Sánchez, América Chávez-Martínez

**Affiliations:** 1Facultad de Zootecnia y Ecología, Universidad Autónoma de Chihuahua, Periférico Fco. R. Almada km 1, Chihuahua 31453, Mexico; ivette@foodsafetycts.com (I.K.R.-R.); arenteria@uach.mx (A.L.R.-M.); rsanchezv@uach.mx (R.S.-V.); jtirado@uach.mx (J.M.T.-G.); esantellano@uach.mx (E.S.-E.); marevalos@uach.mx (M.M.A.-S.); 2Facultad de Ciencias Químicas, Universidad Autónoma de Chihuahua, Circuito Universitario s/n Campus Universitario 2, Chihuahua 31240, Mexico; ngutierrez@uach.mx

**Keywords:** ACE-inhibitory, antioxidant activity, *requeson* cheese, sodium chloride, potassium chloride

## Abstract

*Requeson* cheese is obtained from whey proteins. The production of this cheese is the most economical way to recover and concentrate whey proteins, which is why it is frequently made in some Latin American countries. Four *requeson* cheese treatments were prepared with different concentrations and combinations of salts (sodium chloride and/or potassium chloride) and were conventionally or vacuum packed. Proteolysis, peptide concentration, angiotensin-converting enzyme (ACE) inhibitory and antioxidant (DPPH and ABTS) activities were evaluated over time (one, seven and fourteen days). *Requeson* cheese presented antioxidant and ACE inhibitory activities, however, these values vary depending on salt addition, type of packaging and time of storage. The highest values of antioxidant activity (ABTS) were found in cheese added with 1.5% NaCl and 1.5% (NaCl/KCl, 1:1). Cheese without added salt and vacuum packed presented the highest ACE inhibition percentage at day seven. Therefore, it can be concluded that *requeson* cheese elaborated exclusively of sweet whey, presents antioxidant and ACE inhibition activity. However, for a cheese with ACE inhibitory capacity, it is recommended not to add salts or add at 1% (NaCl) and vacuum pack it. Additionally, for a cheese with antioxidant activity, it is recommended to add salt at 1.5% either NaCl or (1:1) NaCl/KCl and pack it either in a polyethylene bag or vacuum. In conclusion, *requeson* cheese elaborate with 100% sweet whey is a dairy product with antioxidant and ACE inhibition activity, being low in salt and fat.

## 1. Introduction

According to World Health Organization (WHO), cardiovascular diseases are the leading cause of death worldwide. It is estimated that 17.9 million people died in 2019 due to cardiovascular diseases (32% of all registered deaths in the world), low- and middle-income countries accounting for three quarters of this percentage [1]. One way to control cardiovascular diseases is prevention, therefore, it is important to take care with regard to being overweight and avoid obesity, avoid tobacco and alcohol consumption and perform physical activity [1,2]. It is also important to reduce sodium consumption, diets with a high content of this mineral are associated with a higher risk of developing hypertension [3,4]. Likewise, to reduce hypertension, some researchers recommend that in addition to reducing sodium intake, it is important to increase potassium intake [2,5,6].

Meanwhile, there are some drugs that help to treat hypertension. Among these are Angiotensin Converting Enzyme (ACE) inhibitors, although some consumers have chosen to eat healthy foods instead of drug intake to maintain their health [7]. As a part of the renin-angiotensin system, ACE regulates peripheral arterial pressure and catalyzes the conversion of angiotensin I to angiotensin II, which is a powerful vasoconstrictor, causing the contraction of blood vessels and therefore increasing blood pressure [8]. Research has shown that there are foods that contain peptides with antihypertensive activity, and many ACE inhibitor peptides have been isolated from different dairy products [9,10] including milk whey [11]. Many whey protein-derived peptides have been described to demonstrate ACE inhibition activity. Whey protein hydrolysates (WPH) containing peptides derived from the β- lactoglobulin and α-lactalbumin fractions have been shown to demonstrate ACE inhibitory activity [12]. 

Furthermore, peptides derived from whey proteins have showed antioxidant activity, with lactoferrin and β- lactoglobulin being the proteins that most influence the antioxidant properties of milk [13]. Whey is rich in sulfur-containing amino acids (1.7%; [14] and β- lactoglobulin contains all 20 essential amino acids and is a rich source of sulfur. Likewise, whey may contain some free amino acids such as Trp, Phe, Tyr and Cys may exert antioxidant activity [15,16,17]. Sulfhydryls are known free radical scavengers [15]. Amino acids containing sulfur such as methionine and cysteine can be converted into reduced glutathione through redox cycle, which subsequently protect against oxidative damage [18]. Meanwhile, aromatic amino acids (tryptophan, tyrosine and phenylalanine) inhibit a chain reaction of free radicals and have an antioxidant effect [19]. L-histidine has been shown to scavenge both the hydroxyl radical and singlet oxygen (O_2_) in many studies [20]. Lactoferrin is an iron-binding monomeric globular glycoprotein, binding 2 Fe^3+^ ions by each monomer, this iron-binding capacity is likely to contribute to its antioxidant activity [21].

Oxidative stress is one of the main factors responsible for initiating or developing degenerative diseases including cardiovascular diseases [2].

In the last decades, peptides derived from milk whey have been sought that eliminate free radicals since these are precursors of the antioxidant glutathione and exhibit antioxidant activity by suppressing the adverse effects of stress factors [11,13,22,23,24]. 

*Requeson* is a cheese obtained from whey proteins coagulation (by heating in an acidic environment, to promote the collection of curd), which are then salted, drained, and packed [25]. It is considered a fresh cheese, with light flavor and a short shelf-life due to its high water content [26]. Since its production process similar, this is often described as the Spanish version of Ricotta cheese. However, Ricotta cheese is usually elaborated with a mixture of whey and milk, and *requeson* cheese is exclusively elaborated with whey. The production of this cheese is the most economical way to recover and concentrate whey proteins, which is why it is frequently made in some Latin American countries where the infrastructure to dehydrate the whey is still scarce.

During the elaboration of *requeson* cheese, the whey protein is concentrated by the addition of acid, salt and the application of heat, and this process breaks down proteins into peptides and amino acids [19]. Then, these pre-digested forms of whey proteins are effectively absorbed in human gut [27]. At the end, the amino acid composition of the hydrolysate formed mostly depends on the method and hydrolysis conditions, and the number of amino acid bonds that are targeted and broken. The grade of hydrolysis can be calculated to determine the release of peptides and amino acids. The greater the degree of hydrolysis, the smaller the number of amino acids contained in a peptide, resulting in the formation of peptides that provide a bitter taste [18]. Hence, the functional properties of whey proteins depend on processing conditions (heating, acidification, counter ions, ionic strength, storage conditions, etc.) and intrinsic (protein composition, hydrophobicity or hydrophilicity, surface charge, bound flavor ligands, etc.) and extrinsic (pH, salts or ions, water, temperature, oxidation-reduction potential, etc.) parameters [19]. The objective of this study was to evaluate the effect of the packaging type and salt content (type/concentration) on proteolysis, the ACE inhibitory activity and antioxidant activity of *requeson* cheese throughout its shelf-life.

## 2. Materials and Methods

### 2.1. Materials

Whey was obtained from a Chihuahua cheese factory (10 L). This was collected in plastic containers and immediately transported to the laboratory for analysis and use. Whey temperature, pH, and acidity (expressed as percent lactic acid) was evaluated. When necessary, the acidity was adjusted to 0.12% with a 10% solution of calcium hydroxide (J.T. Baker, Mexico). Sodium chloride (La Fina^®^ Salt) was purchased locally in Chihuahua, Mexico, and potassium chloride and calcium chloride were from J.T. Baker, Mexico. Chemicals and reagents were used from the laboratory stock and were of analytical reagent grade. Compounds 2,2′-azino-bis (3-ethylbenzothiazoline-6-sulfonic acid) (ABTS), 2,2-diphenyl-1-picrylhydrazyl (DPPH), Trolox, ammonium salt, potassium persulfate, trichloroacetic acid, sodium dodecyl sulfate (SDS), sodium tetraborate, O-Phthaldialdehyde (OPA), Bradford reagent, bovine serum albumin (BSA) β-Mercaptoethanol, Angiotensin II Converting Enzyme (ACE II), Tris-HCl buffer, Hippuryl-Histidine-Leucine (HHL), trifluoroacetic acid (TFA) and Hippuric acid (HA) were analytical grade and acquired from Sigma-Aldrich (St. Louis, MO, USA). Methanol and water (grade-high performance liquid chromatography) were acquired from J. T. Baker (Mexico City, Mexico). A deionizer was used to obtain deionized water (Barnstead, Thermo Scientific, Waltham, MA, USA).

### 2.2. Requeson Cheese Preparation

Batches of 10 L of sweet whey were used to prepare *requeson* cheese according to Inda’s [19] methodology, with some modifications. First, whey temperature was raised to 96 °C at an approximate rate of 1 °C/min. Upon reaching that temperature, 0.1% citric acid and the corresponding salt were added for each treatment. Subsequently, the temperature was maintained at 96 °C for 10 min, and then allowed to stand for 30 min without stirring. Finally, *requeson* cheese was recovered with blankets, which were hung to eliminate the excess liquid, and then it was reposed for two hours at room temperature. Finally, it was packed and kept refrigerated at 4 °C until further analysis.

### 2.3. Experimental Design

Experimental design contemplated a factorial treatment arrangement, in a completely randomized design and considering the following factors: four salt levels/combination; control (*requeson* cheese without salt), *requeson* cheese with 1% NaCl, 1.5% NaCl and 1.5% (1:1) of NaCl/KCl, two packing types: ordinary retail packaging in a low-density polyethylene bag and a vacuum packaging in a polyethylene/polyamide bag. The percentages and combinations of the salts used to elaborate the *requeson* cheese treatments were chosen according to a previous study, where these were the most sensory accepted by consumers [28].

Analyses were conducted at 1, 7 and 14 days of storage. Peptide concentration, proteolysis, antioxidant (ABTS and DPPH) and ACE inhibitory activities were evaluated over time. For the first three determinations, the water-soluble extract (WSE) of the treatments were obtained and, for the ACE inhibitory activity, the WSE were lyophilized. All analyses were carried out in triplicate. Physicochemical composition, texture and sensory analysis of treatments have been reported previously [28].

### 2.4. Water Soluble Extract (WSE) Preparation

The WSE was obtained based on the methodology described by Torres-Llanez et al. [29]. A sample (30 g of *requeson* cheese) was mixed with distilled water (100 mL), homogenized (Vortex-Ultra-Turrax IKA T18 basic) (S18N-19G, IKA Works Inc., Wilmington, NC, USA) and centrifuged (Avanti Model J-26 XPI Beckman Coulter^®^ centrifuge, Indianapolis, IN, USA) at 20,000× *g* at 4 °C for 30 min. The supernatant was filtered in a grade 1 Whatman ™ filter paper (GE Healthcare, UK) and kept frozen at −20 °C until analysis.

### 2.5. Antioxidant Activity

ABTS assay was slightly modified from the methodology described by Thaipong et al. [30]. The formation of the ABTS•+ radical was carried out by the reaction of potassium persulfate and ABTS. A stock solution of ABTS (7.4 mM) was made (0.0194 g of ABTS, 0.0033 g of potassium persulfate (2.6 mM) and 5 mL of distilled water). The solution was then left in the dark at room temperature for 12 h. For the ABTS working solution, 1 mL of ABTS stock solution was added to 60 mL of methanol until it obtained an absorbance of 1.1 + 0.02. Afterward, 150 µL of the standard (Trolox) or sample and 2900 µL of ABTS working solution were mixed and left in the dark at room temperature for 2 h. Subsequently, the absorbance was measured (734 nm) in a UV spectrophotometer (UV-1800. Shimadzu, Japan). All analyses were performed in triplicate. Antioxidant activity was expressed as equivalent µM Trolox (μM TE/mL). The obtained absorbance was then substituted into the regression equation (y = −0.0013x + 1.0744; r^2^ = 0.9964) derived from the Trolox calibration curve.

The DPPH assay was performed according to Thiapong et al. [30]. A stock solution of 0.6 mM DPPH (0.0240 g DPPH in 100 mL methanol) was first prepared to a concentration of 0.6 mM. This stock solution was stored in the dark and frozen at −20 °C until use. To obtain a working solution, 10 mL of stock solution was mixed with methanol (45 mL) to obtain an absorbance of 1.1 ± 0.02. Next, a 150 µL standard solution (Trolox) or an aliquot of the sample was mixed with a 2900 µL DPPH working solution and the mixture was placed in the dark at room temperature for 3 h. Finally, the absorbance at 515 nm was measured with a UV spectrophotometer (UV1800, Shimadzu, Japan). The measurement was performed three times. Antioxidant capacity was expressed as µM Trolox equivalent (µM TE/mL). The obtained absorbance was then substituted into the regression equation (y = −0.0011x + 0.9747; r^2^ = 0.9892) derived from the Trolox calibration curve.

### 2.6. Proteolysis 

Proteolysis was evaluated according to Church et al. [31]. First, a 20% w/v solution of sodium dodecyl sulfate (SDS, Sigma-Aldrich) was elaborated (0.5 g in 2.5 mL of distilled water). Then, a 100 mM tetraborate sodium solution (Sigma-Aldrich) was prepared (0.953 g in 25 mL of distilled water). Finally, an o-Phthaldialdehyde (OPA, Sigma-Aldrich) solution [0.04 g plus 1 mL of methanol (Sigma-Aldrich) and 100 µL of β-mercaptoethanol (Sigma-Aldrich)] was made. The first two solutions were added to the last one and made up to 50 mL with distilled water and allowed to stand overnight in the dark. Then, 2.5 g were weighed, mixed with 5 mL of trichloroacetic acid (0.75% solution) and filtered through grade 1 Whatman™ filter paper (GE Healthcare, UK). Finally, 1 mL of the prepared solution and 50 μL of the sample filtrate were mixed, shaken, incubated for 2 min, and read at 340 nm with a spectrophotometer (UV1800, Shimadzu, Japan).

### 2.7. Peptide Concentration

It was carried out by Bradford methodology [32]. For this, 2.5 g of *requeson* cheese were mixed with 5 mL of trichloroacetic acid (JT Baker, Center Valley, PA, USA) at 0.75% and homogenized in a porcelain mortar; subsequently, this mixture was filtered with Whatman™ filter paper (GE Healthcare, UK) with 125 mm pore diameter. Then, 0.1 mL of the filtrates or standard were mixed with 1 mL of Bradford reagent (Sigma-Aldrich) and left in the dark for 5 min and then read at a wavelength of 595 nm in spectrophotometer (UV-1800. Shimadzu, Japan). For the calibration curve, bovine albumin standards (Sigma-Aldrich) were prepared in concentrations of 0 to 70 μg/mL to obtain the resulting equation (y = 0.0067x + 0.5423, r^2^ = 0.9902).

### 2.8. ACE Inhibitory Activity

The ACE-inhibitory activity was analyzed by Reverse Phase High Performance Liquid Chromatography (RP-HPLC) as described by Pritchard et al. [33] and Ghassem et al. [34] with slight modifications. One unit of ACE (Sigma-Aldrich) was diluted in 10 mL of 50 mM Tris-HCl (Sigma-Aldrich) (pH = 7.5) with 0.3 M NaCl to obtain a final concentration of 100 mU/mL. Aliquots of 1 mL were frozen at −20 °C until use. 

Samples of the lyophilized filtrates were resuspended in tri-distilled water until reaching a concentration of 50 mg/mL. Next, 50 μL of the resuspended lyophilized filtrates or tri-distilled water (for the control sample) were mixed with 200 μL of 2.5 mM Hipuryl-L-Histidyl-L-Leucine solution (HHL, Sigma-Aldrich) (0.1073 g of HHL in 10 mL with distilled water) and incubated at 37 °C for 10 min.

Subsequently, 10 μL of ACE (100 mU/mL) were added to each sample and incubated again at 37 °C for 30 min with constant shaking. The reaction was stopped with the addition of 250 μL of HCl 1 M. The solution was then filtered (Whatman 30 mm/0.20 μm nylon filters). Finally, 5 μL of samples were injected in a High-Performance Liquid Chromatograph (HPLC) coupled with a UV-visible detector (Shimadzu, Japan). A Discovery^®^ C18 column of 25 cm × 4.6 mm and 5 μm particle size (SUPELCO^®^ Analytical) was used and eluted with water-methanol 50–50% (v/v) acidified with 0.1% trifluoroacetic acid (TFA, Sigma-Aldrich) at a flow rate of 0.4 mL/min and an absorbance of 228 nm. The evaluation of ACE inhibition was based on the comparison between the concentration of hippuric acid (HA, Sigma-Aldrich) in the presence and absence (control sample) of the inhibitor. The area under the curve was determined for each HA peak obtained.

ACE inhibition percentage was determined by the following formula:(1)$$\mathrm{Percentage}\;\mathrm{of}\;\mathrm{inhibition}=\frac{\left(\mathrm{control}-\mathrm{inhibitor}\;\mathrm{sample}\right)}{\left(\mathrm{control}-\mathrm{sample}\;\mathrm{without}\;\mathrm{ACE}\right)}\times 100$$

### 2.9. Statistical Analysis

SAS version 9.1.3 (SAS Institute Inc, Cary, NC, USA) was used for all statistical analysis. Univariate and multivariate analysis of variance, Tukey’s means comparison and correlations tests were conducted with a significance level of 0.05.

## 3. Results and Discussion

### 3.1. Antioxidant Activity

#### 3.1.1. ABTS Methodology

The analysis showed a significant difference (*p* < 0.05) for the triple interaction: treatment, type of packaging, and day (Figure 1). The lowest values were presented by T2 (1% NaCl), either conventional or vacuum packed, at days 1, 7 and 14 with no statistically significant differences between them (*p* > 0.05). The highest values were found in T3 (1.5% NaCl) and T4 [1.5%, (1:1) NaCl/KCl] without any statistically significant differences between them (*p* > 0.05), except on day one and vacuum packed, where T3 had a greater value than T4, 259.41 and 215.44 μM TE/mL, respectively. No statistically significant differences (*p* > 0.05) were observed in the treatments by type of packaging, on the same day, except for T1 (control cheese) on day 1 (*p* < 0.05), in this day, conventional packed (105.43 μM TE) had a lower value than vacuum packed (153.38 µM TE).

#### 3.1.2. DPPH Methodology

Only the day factor showed significance (*p* < 0.05) in the multivariate statistical analysis. In general, a decrease in antioxidant activity was observed with respect to time, regardless of the type of packaging (Figure 2). T1 (control cheese) conventional packed had greater (*p* < 0.05) antioxidant activity at day 14; 120.33 at day 1 versus 85.18 at day 14. T2 (1% NaCl), vacuum packed, presented a higher antioxidant capacity (*p* < 0.05) also on day 1 (121.24) compared to day 14 (85.18).

No literature was found on the antioxidant activity of *requeson* cheese. However, it was observed that values found by the ABTS methodology were higher than those found by DPPH methodology. Several authors have reported the same findings. Floegel et al. [35] found that the antioxidant capacity determined in fruits, vegetables, and beverages, by the ABTS technique report significantly higher values compared to the DPPH technique. Meanwhile, Abadía-García et al. [36] determined antioxidant activity in cottage cheese by the ABTS and DPPH methods; the ABTS values were higher than those reported by the DPPH technique, for example, for the control treatment (cottage cheese without probiotic culture) the value of DPPH on day 14 was approximately 25 µM TE and, for the ABTS methodology it was 300 µM TE. Differences between the two methods could be due to the inequality between the radical structures that can react differently depending on the peptides present in the aqueous extract.

Dairy products and derivates have been found to be antioxidative like fermented milk, yogurt, whey, and cheese [37,38,39,40]. Meanwhile, antioxidant activity of whey protein hydrolysates is greater than in whey protein concentrate [41]. Therefore, a certain degree of hydrolysis is necessary for serum proteins to exert antioxidant activity.

In low-fat cheeses, DPPH values of around 600 μM/g TE have been reported [42]. In addition, Revilla et al. [43] analyzed the antioxidant activity of 224 cheeses made with different types of milk (cow, goat, and sheep) of up to six months of maturation; for cow’s milk, cheeses reported an average value of 7046 μM of Trolox/mg of cheese. These values are higher than the ones reported in this study. These differences could be due to the process and the milk fraction used to elaborate these two dairy products. Cheeses contain peptides with antioxidant activity derived from caseins [38], which is the major protein fraction of milk (80%), compare to *requeson* cheese, which is elaborated with the whey fraction (20%). Generally, the whey portion of milk contains five fractions that altogether make up 85% of the whey protein. These fractions include β-lactoglobulin and α-lactalbumin, glycomacropeptide, immunoglobulins, proteose peptone and serum albumin [44,45]. In addition, the major contributor to the total antioxidant capacity of whole milk is found in the casein fractions [46]. Due to this fact, the antioxidant activity in cheeses elaborated with whole milk is expected to be higher than in *requeson* cheese.

Moreover, it was found that in *requeson* cheese, antioxidant activity decreases with time, contrary to what is reported in cheeses. This may be due to the degree of proteolysis that these products present at the end of its elaboration. An increase in proteolysis during ripening is a well-documented fact for many cheese varieties [47,48,49]. During cheese maturation, intact proteins are hydrolyzed to peptides [50], some of which have antioxidant activity [51]. Moreover, *requeson* is produced from sweet whey, which is coagulated (by heating in acidic medium, to favor the obtaining of curd), salted, drained, and packed [52]. This process causes the denaturation of β-lactoglobulin, a heat labile protein, and the primary whey protein (55–65% of whey proteins), and to a lesser degree, of the α-lactalbumin [53]. The degree of proteolysis that occurs generates peptides with antioxidant activity, however, as proteolysis continues, peptides could be continuously degrading, thus losing their antioxidant activity.

Meanwhile, it is known that vacuum packaging helps to minimize oxidation reactions due to lack of oxygen [54], however, there was no significant difference between types of packaging (*p* > 0.05) on the antioxidant activity of *requeson* cheeses.

### 3.2. Proteolysis

Proteolysis results are shown in Figure 3. In general, proteolysis values were found between 0.389 nm and 0.568 nm. Double interactions treatment/pack, treatment/day and pack/day were significant (*p* < 0.05). T1 (control cheese) presented greater proteolysis at day 14 (*p* < 0.05) in conventional packed (0.568 nm) and vacuum packed (0.518 nm) compared to day one and seven. While proteolysis of treatments T2 (1% NaCl) and T4 [1.5% (1:1) NaCl/KCl] remained constant over time in the two types of packaging (*p* > 0.05). Making a comparison between the T1 (control cheese) with the rest of the treatments that contain salts, it was observed that the salt influenced proteolysis negatively, since a major grade of proteolysis (*p* < 0.05) was observed in *requeson* cheese conventional packed without salt at day 14. Meanwhile, in T1 (control cheese) conventional or vacuum packed, proteolysis was negatively correlated with peptide concentration (*p* < 0.05) (Table 1).

In cheeses, high levels of salt (10%) decreased the speed and/or degree of proteolysis since salt content affects enzymatic activity, due to the high ionic strength exerted by salts that reduces the attractive forces between enzymes and caseins [55]. The same effect probably occurs with whey proteins. The proteolysis of *requeson* cheese could be due to microbial or rennet heat-resistance proteases at low pH (4.5 to 6.3) that are present in the whey [56,57]. Furthermore, most of the activity of the rennet that is added to make the curd during the production of cheddar cheese is recovered in the whey. [58]. Meanwhile, lactic acid bacteria vary in their proteolytic activity and, thus, may contribute to residual protease activity [59]. Generally, the mesophilic culture used to make Chihuahua cheese contains *Lactococcus lactis* and *Streptococcus thermophilus*. Microbiological count of lactic acid bacteria in all treatments were conducted (data do not shown); however, all counts were below detection limits, affirming that proteolysis may be caused by residual heat-resistant proteases found in the whey [60,61,62].

### 3.3. Peptide Concentration

Peptide concentration results are shown in Figure 4. The triple interaction treatment (content and type of salt), packaging type (conventional and vacuum packed) and days of storage of *requeson* cheese was statistically significant (*p* < 0.05). In T1 (control cheese), conventional packaging, a decrease in peptide concentration (*p* < 0.05) was observed on day 14, from an initial concentration of 47.42 to 13.34 μg/mL. For its part, this same treatment, but vacuum packed, showed different behavior; on day one it had a concentration of 46.02, on day seven it increased to 64.53 (*p* < 0.05) and, finally on day 14 it decreased (*p* < 0.05) to 28.91, showing a significant difference over time (*p* < 0.05). T2 (1% NaCl) also showed a significant increase (*p* < 0.05) from day one to day seven, in two types of packaging, and subsequently the peptide concentration decreased on day 14, although the difference from day seven to 14 in vacuum packed was not significant (*p* > 0.05).

In T3 (1.5% NaCl) in a polyethylene bag, on day seven the peptide concentration increased from 34.68 μg to 49.61 μg (*p* < 0.05) and on day 14 it decreased to 44.48 μg without significant difference between day seven and 14 (*p* > 0.05). This same treatment, vacuum-packed, increased from day one to day seven, from 31.70 μg to 42.04 μg (*p* < 0.05) but on day 14 peptide concentration dropped to 29.66 μg, showing a significant difference with day seven (*p* < 0.05).

The same behavior was seen in T4 [1% (1:1) NaCl/KCl] in both packages. Conventional packed, on day one had a peptide concentration of 31.75 μg, this value increase on day seven to 51.15 μg (*p* < 0.05) and, on day 14 it decreased to 26.47 μg, being statistically different from day seven (*p* < 0.05). This same treatment, vacuum packed, presented an increase of peptide concentration from day one (34.23 μg) to day seven (40.65 μg) (*p* > 0.05) and then, on day 14 it decreased significantly to 30.11 μg (*p* < 0.05).

The greatest concentration of peptides was found in the control treatment, vacuum packed, at day seven (*p* < 0.05), and the lowest concentration was detected in the control treatment but conventional packaging at day 14 (*p* < 0.05). As said before, in T1 (control cheese), peptide concentration was negatively correlated with proteolysis (*p* < 0.05) (Table 1) and proteolysis is affected by salt content.

As mentioned before, in *requeson* cheese, the elaboration process causes the denaturation and proteolysis of the β-lactoglobulin and the α-lactalbumin [53]. The extent of the proteolysis depends mainly on the temperature and the time applied during its elaboration. This proteolysis generates peptides, however, as proteolysis continues, peptides could be continuously degrading, thus losing their bioactivity. 

It was also observed that salt inhibits proteolysis; as mentioned before, in cheese, the content of salts affects the enzymatic activity, which is due to the high ionic force that these exert, and that reduces the forces of attraction between enzymes and proteins [55].

Meanwhile, since peptide concentration, including all treatments, was not positively correlated with antioxidant activity, it could be assumed that the antioxidant capacity could be given by free amino acids such as Trp, Phe, Tyr and Cys, since these exert antioxidant activity [15,16,17].

### 3.4. ACE Inhibitory Activity

Figure 5 shows the results of the percentage of ACE inhibition. The triple interaction treatment (content and type of salt), packaging type (conventional and vacuum packed) and days of storage of *requeson* cheese was statistically significant (*p* < 0.05). Treatment one (control), conventional packaging, on day one had an ACE inhibition percentage of 54.70, and on day seven it increased (*p* < 0.05) to 90.46% and, on day 14 it decreased to 82.43% (*p* > 0.05). This treatment, but vacuum packed, presented the highest ACE inhibition percentage at day seven (126.37%) (*p* < 0.05), then at day 14 this value decreases to 82.02% (*p* < 0.05).

The T2 (1% NaCl), conventional packed, on day one had an ACE inhibition value of 76.58%, on day seven it increased to 99.96% (*p* < 0.05) and finally, on day 14 it decreased to 95.70% (*p* > 0.05). This treatment vacuum packaged had the same behavior; on day one the ACE inhibition value was 66%, on day seven it increased to 111.19% (*p* < 0.05) and on day 14 it decreased 88.04% (*p* < 0.05).

The ACE inhibition percentage in vacuum packed T3 (1.5% NaCl) increased with respect to time from 47.52%, to 81.53% and to 89.04%, making values of day one significantly different to days seven and 14 (*p* < 0.05). This treatment, conventional packaging, presented the same behavior, having values on day one (71.01%) significantly different to days seven (78.92%) and 14 (88.94%) (*p* < 0.05).

Finally, T4 [1.5% (1:1) NaCl/KCl], conventional packed, on day one showed a 58.73% of ACE inhibition, on day seven it increased to 80.74% and on day 14 it increased to 85.64%, having values on day one significantly different to days seven and 14 (*p* < 0.05). Furthermore, this treatment, vacuum packed, showed an ACE inhibition value of 38.56% on day one and, on day seven it increased to 48.37% and on day 14 it increased to 90.08%; this last value was statistically different (*p* < 0.05) from the one found on days one and seven. Donkor et al. [63] analyzed yogurt samples with probiotics and obtained the same non-linear behavior with respect to time in the percentage of ACE inhibition; their values were between 70% and 90% during a storage period of 28 days in refrigeration. The highest ACE inhibition values were found in T1 and T2 at day seven, vacuum packed; T1 (control) with an inhibition percentage of 126.37% and T2 (1% NaCl) with 111.19% (*p* > 0.05). In these two treatments, ACE inhibition was positively correlated with peptide concentration (*p* < 0.05) (Table 1 and Table 2). The lowest values of the inhibition percentage were found in treatments three (1.5% NaCl) and four (1.5% NaCl/KCl 1:1), vacuum packed, at day one (*p* > 0.05); T3 with a value of 47.5% and T4 with 38.5%, both on day one and vacuum packed (*p* > 0.05). In these two treatments, ACE inhibition was negatively correlated with antioxidant activity (ABTS) (*p* < 0.05) (Table 3 and Table 4).

It has been recognized that peptides derived from the major whey proteins, i.e., α-lactalbumin and β-lactoglobulin in addition to bovine serum albumin (BSA) exert ACE inhibition activity [60]. Protein breakdown in whey protein products and foods containing whey protein may be caused by residual plasmin or heat stable proteases produced by starter cultures [60,61,62]. Meanwhile, studies have concluded that there is no relationship between the degree of hydrolysis and ACE inhibitory activity and concluded that enzyme specificity, rather than the degree of hydrolysis, determined the development of ACE inhibitory peptides [12,64]. To date, no studies are available on the ACE inhibitory activity during the shelf-life of *requeson* cheese elaborated exclusively with cheese whey. Madureira et al. [65] evaluated this parameter in *requeson* cheese elaborated with 90/10% cheese whey/whole milk, inoculated with dairy cultures (*Lactobacillus casei* and *Bifidobacterium animalis* strains), and found that *requeson* cheese at 21 days of storage showed the highest ACE inhibitory activity, with values that were 103 µg mL^−1^ and 141 µg mL^−1^, respectively (values expressed as IC_50_).

### 3.5. Correlations

Table 5 shows the correlation coefficients of peptide concentration, proteolysis, DPPH, ABTS and ACE inhibitory activity in *requeson* cheese treatments. Proteolysis was negatively correlated with antioxidant activity by both methods DPPH and ABTS, −0.253 (*p* = 0.031) and −0.246 (*p* = 0.037), respectively; and peptide concentration −0.437 (*p* = 0.001). In addition, peptide concentration was negatively correlated with antioxidant activity (ABTS) −0.256 (*p* = 0.030). Finally, ACE inhibition was negatively correlated with antioxidant activity by both methods DPPH and ABTS, −0.333 (*p* = 0.004) and −0.377 (*p* = 0.001), respectively; and positively correlated with peptide concentration 0.364 (*p* = 0.001). According to these results, a higher degree of proteolysis less peptide concentration and less antioxidant activity. This result affirms what was mentioned previously with regard to ACE inhibition; the more peptide concentration, the more ACE inhibition activity, and the less antioxidant activity. This finding suggests two assumptions: the first is that the antioxidant activity could also be given by free amino acids such as Trp, Phe, Tyr and Cys, since these exert antioxidant activity; and the second suggests that most of the peptides are formed during the elaboration of *requeson* cheese, thus the less proteolysis the more peptides with either antioxidant or ACE inhibition activity remain intact; however, if proteolysis continues, peptides could be continuously degrading until they lose their bioactivity. To verify this, it would be necessary to analyze the content of free amino acids and carry out peptide profiles during the production and shelf life of the cottage cheese.

## 4. Conclusions

The *requeson* cheese presented peptides with antioxidant and ACE inhibitory activity and these activities vary depending on the percentage of salt addition, type of packaging and time of storage. The highest values of antioxidant activity (ABTS) were found in T3 (1.5% NaCl) and T4 [1.5%, (1:1) NaCl/KCl]. This same activity, but conducted using the DPPH methodology, showed a decrease in activity with respect to time, regardless of the type of packaging and treatment. Finally, T1 (control), vacuum packed, presented the highest ACE inhibition percentage at day seven. These bioactivities are due to the formation of peptides during *requeson* cheese elaboration and could also be due to residual enzymatic activity. Therefore, it would be advisable to study the presence of enzymes in *requeson* cheese. Therefore, it can be concluded that *requeson* cheese elaborated exclusively of sweet whey, presented antioxidant and ACE inhibition activity, and that these activities vary depending on storage time, type of packaging and type and concentration of the added salt.

## Figures and Tables

**Figure 1 foods-11-01264-f001:**
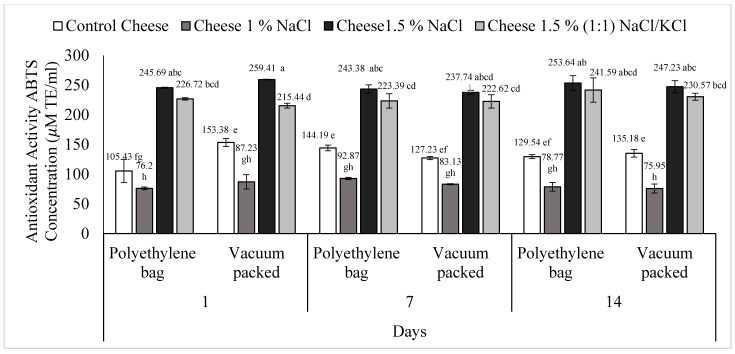
Antioxidant activity (ABTS˙ method) of *requeson* cheese with different salt type and concentration treatments, conventional and vacuum packaging. ^a–h^ Values with a different superscript indicate significant statistical difference (*p* < 0.05).

**Figure 2 foods-11-01264-f002:**
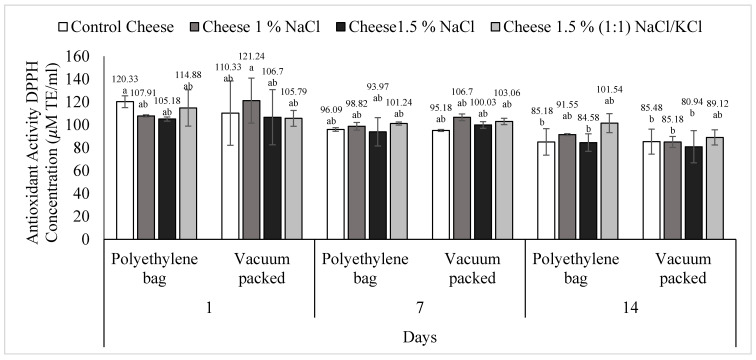
Antioxidant activity (DPPH˙ method) of *requeson* cheese with different salt type and concentration treatments, conventional and vacuum packaging. ^a,b^ Values with a different superscript indicate significant statistical difference (*p* < 0.05).

**Figure 3 foods-11-01264-f003:**
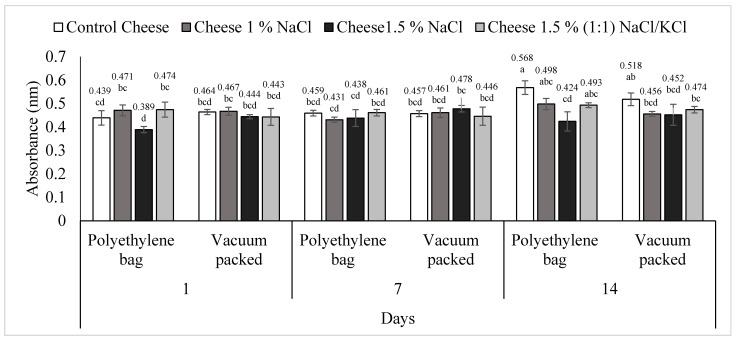
Proteolysis of *requeson* cheese with different salt type and concentration treatments, conventional and vacuum packaging. ^a–d^ Values with a different superscript indicate significant statistical difference (*p* < 0.05).

**Figure 4 foods-11-01264-f004:**
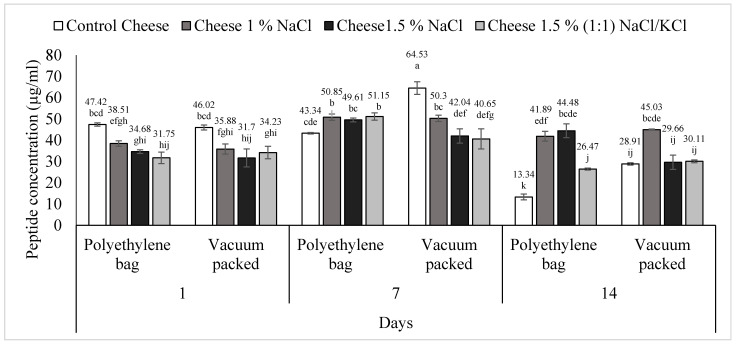
Peptide concentration of *requeson* cheese with different salt type and concentration treatments, conventional and vacuum packaging. ^a–k^ Values with a different superscript indicate significant statistical difference (*p* < 0.05).

**Figure 5 foods-11-01264-f005:**
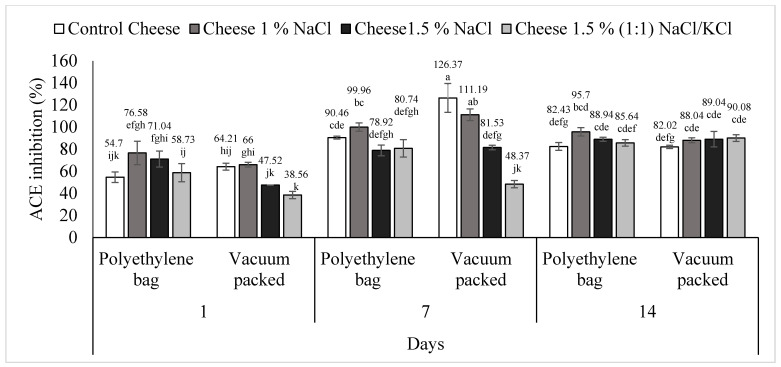
ACE-inhibition of *requeson* cheese with different salt type and concentration treatments, conventional and vacuum packaging. ^a–k^ Values with a different superscript indicate significant statistical difference (*p* < 0.05).

**Table 1 foods-11-01264-t001:** Correlation between response variables of T1 *requeson* cheese (Control).

	Conventional Packed				
	ABTS	DPPH	Proteolysis	PC	ACE
ABTS	1	−0.516	0.314	−0.200	0.840
*p*-value		0.154	0.409	0.605	0.004
DPPH		1	−0.699	0.746	−0.794
*p*-value			0.036	0.020	0.010
Proteolysis			1	−0.940	0.398
*p*-value				0.000	0.288
PC				1	−0.394
*p*-value					0.292
ACE					1
	Vacuum Packed				
ABTS	1	0.547	−0.136	−0.268	−0.832
*p*-value		0.127	0.727	0.4847	0.005
DPPH		1	−0.371	0.236	−0.236
*p*-value			0.325	0.539	0.539
Proteolysis			1	−0.801	−0.331
*p*-value				0.009	0.383
PC				1	0.685
*p*-value					0.041
ACE					1

ABTS˙ and DPPH˙ = Antioxidant activity, PC = Peptide concentration, ACE = Angiotensin converting enzyme. *p*-value = Level of significance of the correlation.

**Table 2 foods-11-01264-t002:** Correlation between response variables of T2 *requeson* cheese (1% NACI).

	Conventional Packed				
	ABTS	DPPH	Proteolysis	PC	ACE
ABTS	1	−0.178	−0.718	0.836	0.569
*p*-value		0.645	0.029	0.005	0.109
DPPH		1	−0.285	−0.323	−0.689
*p*-value			0.457	0.39	0.039
Proteolysis			1	−0.664	−0.015
*p*-value				0.050	0.96
PC				1	0.724
*p*-value					0.027
ACE					1
	Vacuum Packed				
ABTS	1	0.177	0.233	−0.167	−0.2060
*p*-value		0.648	0.546	0.667	0.594
DPPH		1	0.169	−0.512	−0.297
*p*-value			0.663	0.158	0.436
Proteolysis			1	−0.139	−0.314
*p*-value				0.72	0.409
PC				1	0.939
*p*-value					0.000
ACE					1

ABTS˙ and DPPH˙ = Antioxidant activity, PC = Peptide concentration, ACE = Angiotensin converting enzyme. *p*-value = Significance level of the correlation.

**Table 3 foods-11-01264-t003:** Correlation between response variables of T3 *requeson* cheese (1.5% NACl).

	Conventional packed				
	ABTS	DPPH	Proteolysis	PC	ACE
ABTS	1	−0.164	0.026	0.023	0.269
*p*-value		0.672	0.945	0.952	0.483
DPPH		1	−0.626	−0.593	0.715
*p*-value			0.071	0.091	0.030
Proteolysis			1	0.747	0.353
*p*-value				0.020	0.350
PC				1	0.461
*p*-value					0.211
ACE					1
	Vacuum packed				
ABTS	1	0.349	−0.754	−0.610	−0.649
*p*-value		0.355	0.018	0.080	0.050
DPPH		1	−0.347	−0.032	−0.463
*p*-value			0.358	0.934	0.209
Proteolysis			1	0.529	0.194
*p*-value				0.142	0.616
PC				1	0.115
*p*-value					0.766
ACE					1

ABTS˙ and DPPH˙ = Antioxidant activity, PC = Peptide concentration, ACE = Angiotensin converting enzyme. *p*-value = Significance level of the correlation.

**Table 4 foods-11-01264-t004:** Correlation between response variables of T4 *requeson* cheese (1.5% NACI/KCI).

	Conventional packed				
	ABTS	DPPH	Proteolysis	PC	ACE
ABTS	1	0.068	0.268	−0.433	0.184
*p*-value		0.860	0.485	0.243	0.634
DPPH		1	−0.026	−0.289	−0.366
*p*-value			0.946	0.450	0.331
Proteolysis			1	−0.561	0.178
*p*-value				0.115	0.645
PC				1	0.109
*p*-value					0.780
ACE					1
	Vacuum packed				
ABTS	1	−0.489	0.233	−0.156	0.665
*p*-value		0.180	0.545	0.687	0.050
DPPH		1	−0.594	0.617	−0.875
*p*-value			0.091	0.076	0.002
Proteolysis			1	−0.608	0.487
*p*-value				0.081	0.183
PC				1	−0.604
*p*-value					0.084
ACE					1

ABTS˙ and DPPH˙ = Antioxidant activity, PC = Peptide concentration, ACE = Angiotensin converting enzyme. *p*-value = Significance level of the correlation.

**Table 5 foods-11-01264-t005:** Correlation between the response variables of *requeson* cheese.

	ABTS	DPPH	Proteolysis	PC	ACE
ABTS˙	1	−0.0868	−0.2461	−0.2546	−0.3339
*p*-value		0.4683	0.0371	0.0309	0.0042
DPPH˙		1	−0.2536	0.0853	−0.3779
*p*-value			0.0316	0.4761	0.0011
Proteolysis			1	−0.4375	0.1300
*p*-value				0.0001	0.2763
PC				1	0.3644
*p*-value					0.0016
ACE					1
*p*-value					

ABTS˙ and DPPH˙ = Antioxidant activity, PC = Peptide concentration, ACE = Angiotensin converting enzyme. *p*-value = Significance level of the correlation.

## Data Availability

The datasets generated during and/or analyzed during the current study are available from the corresponding author on reasonable request.

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
