# Peer review of "Effect of Packaging and Salt Content and Type on Antioxidant and ACE-Inhibitory Activities in Requeson Cheese"

_foods, 2022, doi:10.3390/foods11091264_

Round 1

Reviewer 1 Report

Regarding the manuscript, I would like to acknowledge novelty concerning Requeson cheese analysis. Manuscript gives important insight in possible health benefits of the consumption of the Requeson cheese. However, I have noticed some shortcoming, especially in the part of the discussion of the results.

Suggestions to consider are given in the attached document.

Author Response

Dear reviewer, The authors have taken into account each of your comments and have made the corrections, we hope that these are appropriate. thanks for your comments.

Reviewer 2 Report

The manuscript describes the effect of packaging and salt content on a less known cheese, Requeson cheese, on the basis of antioxidant activity and angiotensin-inhibitory enzyme. The exploitation of whey protein to prepare this type of cheese is very important to support the food waste application and the wellfare of feeding systems in the near future. In addition, preparation of such a cheese supports further the circular economy. As the authors showed, the properties of this type of cheese are affected by the packaging systems, added salt, and storage time. The manuscript is a vovel research study. It is well written and organized.  In fact, scarce literature is available for this topic.The tables and figures are of very good quality and statistical nalaysis has been done. My comments for authors are enclosed in the attached pdf with yellow colour and inserted comments.

Based on these comments, I suggest a minor revision.

Author Response

Dear reviewer,

The authors have taken into account each of your comments and have made the corrections (this are found in the corrected manuscript), we hope that these are appropriate.

Thanks for your comments.

Reviewer 3 Report

This article is interesting, well designed and well presented.

My comments:

Ln 98: correct (were from (JT Baker,Mexico) chemical…………)

Ln 107: delete (And)

Ln130: delete (1,7, and 14 days)

Ln 223 rephrase title to be (Antioxidant Activity)

The discussion part of antioxidant activity need to be more in depth in relation to the treatment and antioxidant activity

Ln281; correct (protease peptone)

Ln282, correct (milk it is found)

Ln304: show the units for presented values (0.568) and 0.518

L305:  4 should be T4

Ln 327: correct the format of scientific names (Lactococcus lactis and Streptococcus thermophiles)

Ln329-330 please add reference for such protease resistant to such high conditions of tepm, acid, time

Discussion and explanations for the results are required for the peptide concentration and treatments effect

Ln 427-428: correct the format of scientific names  

The authors need to clarify and highlight the final recommendation showing the best conditions for keeping requeson cheese in term of preparation and preservation conditions..plrase clarify in abstract and conclusion sections

Carefully check references especially for scientific names to be corrected (see for example Ln528, Ln539, Ln541,.etc)

Author Response

Dear reviewer,

The authors have taken into account each of your comments and have made the corrections (this are found in the corrected manuscript), we hope that these are appropriate.

Thanks for your comments

Reviewer 4 Report

Please find attached my comments and sugestions.

Author Response

(The authors gave the same response as above.)

Round 2

Reviewer 1 Report

The Authors answered all the sugested points for consideration given in the first review.

Author Response

The authors appreciate your comments as they undoubtedly managed to improve the quality of the manuscript.

Reviewer 4 Report

Please find attached my comments. Since most of the corrections are related to the quality of english presentation, I think that I can not help authors on this aspect. Hence, I recommended the revision of the manuscript by an native english speaker.

Author Response

Dear reviewer,

The authors appreciate your comments as they undoubtedly managed to improve the quality of the manuscript. All the suggested changes were made, except for one in which the authors did not know where to make it.
